# Hepatic Venous Occlusion Type of Budd–Chiari Syndrome versus Pyrrolizidine Alkaloid-Induced Hepatic Sinusoidal Obstructive Syndrome: A Multi-Center Retrospective Study

**DOI:** 10.3390/jpm13040603

**Published:** 2023-03-30

**Authors:** Yaru Tong, Ming Zhang, Zexue Qi, Wei Wu, Jinjun Chen, Fuliang He, Hao Han, Pengxu Ding, Guangchuan Wang, Yuzheng Zhuge

**Affiliations:** 1Department of Gastroenterology, Nanjing Drum Tower Hospital, The Affiliated Hospital of Nanjing University Medical School, Nanjing 210008, China; 2Department of Gastroenterology, Shandong Provincial Hospital Affiliated to Shandong First Medical University, Jinan 250021, China; 3Department of Gastroenterology, The First Affiliated Hospital of Wenzhou Medical University, Wenzhou 325015, China; 4Hepatology Unit, Department of Infectious Diseases, Nanfang Hospital, Southern Medical University, Guangzhou 510515, China; 5Liver Disease Center, Beijing Friendship Hospital Affiliated to Capital Medical University, Beijing 100050, China; 6Department of Ultrasound, Nanjing Drum Tower Hospital, The Affiliated Hospital of Nanjing University Medical School, Nanjing 210008, China; 7Department of Intervention, The First Affiliated Hospital of Zhengzhou University, Zhengzhou 450052, China

**Keywords:** Budd–Chiari syndrome, hepatic sinusoidal obstructive syndrome, Doppler ultrasonography

## Abstract

(1) Background: Hepatic venous occlusion type of Budd–Chiari syndrome (BCS-HV) and pyrrolizidine alkaloid-induced hepatic sinusoidal obstructive syndrome (PA-HSOS), share similar clinical features, and imaging findings, leading to misdiagnoses; (2) Methods: We retrospectively analyzed 139 patients with BCS-HV and 257 with PA-HSOS admitted to six university-affiliated hospitals. We contrasted the two groups by clinical manifestations, laboratory tests, and imaging features for the most valuable distinguishing indicators.; (3) Results: The mean patient age in BCS-HV is younger than that in PA-HSOS (*p* < 0.05). In BCS-HV, the prevalence of hepatic vein collateral circulation of hepatic veins, enlarged caudate lobe of the liver, and early liver enhancement nodules were 73.90%, 47.70%, and 8.46%, respectively; none of the PA-HSOS patients exhibited these features (*p* < 0.05). DUS showed that 86.29% (107/124) of patients with BCS-HV showed occlusion of the hepatic vein, while CT or MRI showed that only 4.55%(5/110) patients had this manifestation (*p* < 0.001). Collateral circulation of hepatic veins was visible in 70.97% (88/124) of BCS-HV patients on DUS, while only 4.55% (5/110) were visible on CT or MRI (*p* < 0.001); (4) Conclusions: In addition to an established history of PA-containing plant exposure, local hepatic vein stenosis and the presence of collateral circulation of hepatic veins are the most important differential imaging features of these two diseases. However, these important imaging features may be missed by enhanced CT or MRI, leading to an incorrect diagnosis.

## 1. Introduction

Although liver vascular diseases affect less than 5 in 10,000 patients, they collectively cause many rare but important liver-related conditions [1]. Budd–Chiari syndrome (BCS), especially the hepatic vein occlusion type of BCS (BCS-HV), and hepatic sinusoidal obstruction syndrome induced by pyrrolizidine alkaloids (PA-HSOS) are uncommon and quite difficult to distinguish [2,3] owing to similar clinical and imaging characteristics such as abdominal distension, hepatomegaly, ascites, mild to moderate abnormal liver function, and uneven enhancement of the liver on enhanced computed tomography (CT) and magnetic resonance imaging (MRI) [3,4,5]. However, according to the current classification of vascular liver disease, BCS-HV and HSOS are considered to be unique diseases due to their different etiologies, pathologies, and their natural clinical course [1,6]. Up to now, some studies have compared BCS and PA-HSOS, however, they have not adequately solved the problem of differential diagnosis between hepatic venous occlusion types of BCS and PA-HSOS [7,8,9]. In this study, we focused on comparing the differences in the clinical, laboratory, and imaging features between BSC-HV and PA-HSOS to identify critical features and diagnostic methods that can improve the differential diagnosis of the two diseases.

## 2. Materials and Methods

### 2.1. Clinical Data

A total of 150 BCS-HV and 306 PA-HSOS patients from multiple centers were retrospectively enrolled in our study between January 2014 and November 2021. The participating hospitals included the Affiliated Drum Tower Hospital of Nanjing University Medical School; the Shandong Provincial Hospital Affiliated to Shandong First Medical University; the First Affiliated Hospital of Zhengzhou University, Nanfang Hospital, Southern Medical University; The First Affiliated Hospital of Wenzhou Medical University; and the Beijing Friendship Hospital Affiliated to Capital Medical University. We included patients aged 12–80 years old with a diagnosis of BCS-HV or PA-HSOS according to the disease diagnostic criteria [10,11,12,13]. The exclusion criteria included: malignancy, severe cardiopulmonary disease, cirrhosis or liver diseases of other causes, or a large amount of missing clinical data.

The diagnosis of BCS was made according to the 2016 EASL clinical practice guidelines [1]. PA-HSOS diagnosis conforms to the “Nanjing criteria” of relevant consensus for PA-HSOS [13], and its clinical components include: (1) definite medical history of PA-containing plant exposure (the patient ingested some plants containing PA reported in previous researches, such as panax Gynura japonica.); (2) abdominal distension and/or liver pain, hepatomegaly, and ascites; (3) serum total bilirubin (STB) increased or abnormal serum markers of liver function; (4) typical features on enhanced CT or MRI (for example, diffuse hepatomegaly, ascites, and plain film scanning showed a heterogeneous decrease in the density of the liver; enhancement on CT or MRI was characterized by a map-like or mottled non-uniform appearance in the venous phase and balanced phase; hepatic vein lumen was narrow or blurred; the hepatic segment of the inferior vena cava was compressed and thinner; MRI results were similar to CT results). In the study, the grading criteria for ascites are as follows: ① the mild ascites can only be detected by abdominal ultrasound; ② the maximum depth of ascites detected by supine ultrasound is less than 3 cm; ③ the moderate to severe ascites were defined as ≥3 cm and 10 cm, respectively [14,15]. The diagnostic criteria of liver cirrhosis in this study are as follows: ① the appearance of the liver is not smooth; ② liver fissure is widened; ③ the patient who has received liver biopsy in the past which was confirmed as cirrhosis.

### 2.2. Enhanced CT and MRI

All computed tomography (CT) examinations were performed with one of the following three scanners: 64-detector spiral row scanner (Somatom Definition AS, Siemens, Munich, Germany), 32-detector row dual-source CT scanner (Somatom Definition, Siemens, Germany), 320-detector row dynamic volume CT (Aquilion ONE 640, Toshiba, Tokyo, Japan). Contrast-enhanced CT was performed after the injection of contrast medium (iohexol, GE Healthcare Co., Ltd., Beijing, China; lopromide, Bayer Healthcare Co., Ltd., China; Loversol, Hengrui Medicine Co., Ltd., Jiangsu, China). MRI data were acquired using a 3.0T MR-scanner (Achieva TX; Philips Medical Systems, Eindhoven, the Netherlands) with an eight-channel head coil. Structural images were acquired with high-resolution T1-weighted three-dimensional fast field echo structural scans. All CT and MRI images were reviewed in consensus by two experienced observers blinded to clinical data. Finally, 130 BCS-HV patients received the above examinations, including 126 patients receiving enhanced CT and 4 patients receiving enhanced MRI. In addition, 175 patients in the PA-HSOS group received enhanced CT and 2 patients received enhanced MRI. On contrast-enhanced CT or MRI, the local occlusion and stenosis of the hepatic vein after the injection of contrast agent can be manifested as a filling defect of the contrast agent. CT and MRI images and reports of all patients were collected retrospectively.

### 2.3. Doppler Ultrasonography

All examinations were performed by an attending physician with at least 3 years of experience. Doppler ultrasound (DUS) images were acquired with a Siemens Acuson S2000 system (Siemens Medical Solutions, Mountain View, CA, USA), equipped with a C-1 curved array transducer. All patients fasted for 6–8 h before the examination and lay supine with their right arm raised above their head to expand the intercostal space. The portal vein diameter and peak velocity were measured at 1–2 cm from the junction of the superior mesenteric vein and splenic vein; the splenic vein diameter and peak flow velocity were measured at 1–2 cm near the splenic hilum. All measurements were made three times by the same operator. The local stenosis or occlusion of hepatic veins can be seen in DUS with the following characteristics: ① hypoechoic substances fill the veins, ② venous stenosis with upstream dilation, ③ fibrous hyperechoic cords replace the veins, and ④ intraluminal thrombus. The presence of at least one of these imaging features is sufficient for the diagnosis of BCS-HV. The DUS images and reports of all patients were collected retrospectively.

### 2.4. Digital Subtraction Angiography

The methods of digital subtraction angiography (DSA) involved in this study include hepatic vein pressure gradient measurement (HVPG), transjugular intrahepatic portosystemic shunt (TIPS), hepatic vein balloon dilation and hepatic vein stent implantation. All patients were monitored for vital signs during operation and were in a supine position. DSA operations are performed by a team of experienced doctors under local anesthesia using X-ray. The hepatic veins of all patients were observed by placing a catheter into the jugular vein through the hepatic vein and then injecting contrast agent. The portal vein was observed by placing a catheter through the femoral vein for indirect portography.

### 2.5. Statistics

All statistical analyses were performed using IBM SPSS Statistics 22.0 (Armonk, NY, USA). Normal continuous variables are described as means  ±  standard deviation (SD) and were compared by independent samples *t*-tests. Non-normal continuous variables are described as medians and interquartile ranges (IQR) and were compared with the Mann–Whitney U test. Categorical variables were described in terms of ratios or percentages and were compared using Chi-squared tests. A *p*-value of <0.05 was used to determine significant differences.

## 3. Results

### 3.1. Clinical Presentation

139 BCS-HV and 257 PA-HSOS patients (Figure 1) met all study criteria and were included in our analyses. Two patients with PA-HSOS were not included in the study due to lack of a large number of laboratory and imaging results. The demographic and clinical characteristics of the cohort are summarized in Table 1. The mean BCS-HV patient age was 43.00 ± 13.08 years and 62.75 ± 9.68 for PA-HSOS (*p* < 0.05). There was no significant difference between patient sex (male 48.20% in BCS-HV vs. 51.40% in PA-HSOS, *p* = 0.549). Patients with PA-HSOS had a shorter duration of illness (median of 1 month with 96.90% of ≤6 months) than those with BCS-HV (*p* < 0.05). Except for lower limb edema, the main clinical features, including abdominal distension, abdominal pain, ascites, abdominal varicose veins, and jaundice, overlapped between the two groups but their incidence was significantly different. Most importantly, 210 (81.71%) patients with PA-HSOS had a clear medical history of taking herbs containing pyrrole alkaloids, while BCS-HV patients had no similar medical history.

### 3.2. Laboratory Tests

The results of laboratory tests are provided in Table 2. In this study, we observed that patients with BCS-HV had significantly lower white blood counts (WBC) and hemoglobin (Hb) than patients with PA-HSOS (*p* < 0.05). The platelet count level in the BCS-HV group was significantly higher than in the PA-HSOS group (*p* < 0.05). The median levels of liver enzymes (alanine aminotransferase, aspartate aminotransferase, and alkaline phosphatase) and STB in the BCS-HV patients were within the normal ranges and significantly lower than in the PA-HSOS group. Although there was some impairment in the hepatic synthetic capacities in both groups, those in the BCS-HV group were less severe with higher albumin levels (36.45 ± 5.64 g/L) compared to PA-HSOS patients (32.70 ± 3.52 g/L) (*p* < 0.05). Coagulation function was also poorer in PA-HSOS patients with a significantly prolonged prothrombin time (PT) and activated partial thromboplastin time (APTT) (*p* < 0.05). Child–Pugh scores were significantly lower in BCS-HV than in PA-HSOS patients. Serum creatinine was significantly higher in PA-HSOS than in the BCS-HV group (*p* < 0.05).

### 3.3. Imaging Findings

Table 3 illustrates the image features found with DUS, CT, MRI, and digital subtraction angiography (DSA) in our cohort. Hepatomegaly with heterogeneous density can be seen on all imaging examinations in the two diseases. However, we found major differences in the imaging presentation between the two groups of patients. Portal vein thrombosis (PVT) was confirmed in six of the BCS-HV patients (4.4%, 6/136), four of which were accompanied by imaging manifestations of liver cirrhosis (including uneven or serrated liver surface and small liver volume), while the PA-HSOS patients showed a significantly higher prevalence of PVT (11.7%, 30/257, *p* = 0.018). The proportion of patients with imaging manifestations of cirrhosis was 46.76% (65/139) in the BCS-HV group and 1.95% (5/257) in PA-HSOS (*p* < 0.001).

The main imaging features of BCS-HV included collateral circulation between hepatic veins (73.90%), esophageal varicose veins (64.00%), splenomegaly (63.80%), hepatomegaly (56.20%), stenosis of the hepatic segmental inferior vena cava (55.20%), and caudate lobe enlargement (47.70%). The main imaging features of PA-HSOS were hepatomegaly (67.10%) and stenosis of the hepatic segmental inferior vena cava (44.20%). In addition, 78.46% (102/130) of BCS-HV patients and 92.66% (164/177) of PA-HSOS patients showed patchy enhancement of the liver on contrast-enhanced CT (*p* < 0.001). It is notable that, unlike BCS-HV, PA-HSOS patients rarely showed collateral circulation of hepatic veins, caudate lobe enlargement, esophageal varicose veins, and splenomegaly. The diameters of the hepatic veins, the portal vein, and the splenic vein were significantly larger in the BCS-HV group than in the PA-HSOS group. Peak flow velocities in the portal and splenic veins were significantly slower in the PA-HSOS group than in BCS-HV patients. Figure 2 shows the typical imaging findings of the two groups.

According to the data shown in Table 3, there are marked significant differences between the two groups in the prevalence of local stenosis or the occlusion of hepatic veins, communicating branches between hepatic veins, enlargement of the caudate lobe, and early liver nodule enhancement (*p* < 0.05). Further, we wanted to test the difference between ultrasound and enhanced CT or MRI in their ability to identify these important imaging features. The data in Table 4 show that Doppler ultrasound and enhanced CT or MRI had different capacities for assisting in the identification of these features. In Doppler ultrasonography of BCS-HV patients, 86.29%(107/124) of patients were observed to show local hepatic vein stenosis or obstruction, while only 4.55% (5/110) were identified on enhanced CT or MRI (*p* < 0.001). In addition, 70.97% (88/124) of patients with BCS-HV showed communicating branches between hepatic veins on ultrasound, but only 4.55% (5/110) were seen on enhanced CT or MRI (*p* < 0.001). Both hepatic caudate lobe enlargement and early enhancing nodules were observed only on enhanced CT or MRI. In patients with PA-HSOS, neither ultrasound nor enhanced CT or MRI demonstrated collateral circulation of hepatic veins, hepatic caudate lobe enlargement, or early enhancing nodules. Figure 3 shows the Doppler ultrasound image data of the two groups of patients.

## 4. Discussion

BCS is characterized by the obstruction of hepatic venous outflow anywhere from the small hepatic veins to the junction of the inferior vena cava (IVC) and the right atrium [1,16]. Based on the anatomical location of the obstruction, BCS can be subdivided into an inferior vena cava obstruction type (BCS-IVC), a hepatic vena obstruction type (BCS-HV), or a combination type (BCS-C) [17,18]. The differential diagnosis between BCS-IVC or BCS-C type and HSOS is not difficult, because the former involves the inferior vena cava, leading to IVC blood flow impairment and the establishment of collateral circulation, especially the expansion of the azygos and semiazygos veins, as well as lower limb edema and pigmentation, which greatly assist clinical diagnosis [19,20]. The diagnosis of HSOS associated with hematopoietic stem cell transplantation (HSCT-HSOS) is also relatively straightforward because it usually occurs in high-risk patients who are treated with myeloablative therapy before HSCT implantation, and physicians have a high level of vigilance for liver injury after myeloablative conditioning [5,21,22,23]. It is recognized that the inclusion of heterogeneous BCS and HSOS populations in research introduces myriad variables that can make it difficult to properly identify key factors that uniquely characterize each disease, thus weakening our ability to distinguish BCS-HV and PA-HSOS. As such, to identify a characteristic set of distinguishing features, we only focused on patients with BCS-HV and PA-HSOS in this study.

PA-HSOS is likely to be missed or misdiagnosed due to its sporadic nature, lack of specific symptoms and signs, and an uncertain or unavailable PA exposure history. In 2019, an expert consensus on the diagnosis and treatment of PA-HSOS was published, proposing the “Nanjing Criteria” for the diagnosis of PA-HSOS [10]. In 2021, we further demonstrated that the “Nanjing Criteria” displays satisfactory sensitivity and specificity for diagnosing PA-HSOS. It was in this validation study that we found that PA-HSOS and BCS, especially BCS-HV, were most likely to be confused among patients whose history of PA exposure could not be established [2]. Both PA-HSOS and BCS-HV are rare vascular diseases of the liver [2,24,25] with overlapping clinical features and are thus difficult to distinguish. As such, there is a critical need to identify criteria that will help to differentiate the two diseases.

To the best of our knowledge, although there have been studies comparing BCS and HSOS, there are still some shortcomings in these studies. D Feng et al. summarized the clinical manifestations and CT features of PA-HSOS patients and compared them with BCS. However, the sample size in this study was small, with only 16 patients of all types of BCS [7]. Y Song et al. compared the laboratory examination and CT features of PA-HSOS and BCS patients, emphasizing the clinical, imaging, and prognostic features of PA-HSOS patients. Unfortunately, the shortcoming of this study lies in the inclusion of a heterogeneous set of BCS patients and a lack of ultrasound data [8,9]. In our study, however, we observed that ultrasound is critical to the differential diagnosis of the two diseases.

Our study shows that the age of onset of BCS-HV is lower than PA-HSOS. In China, older people are more likely to receive plant-based drugs than young adults for health care or the management of chronic pain diseases. Only 81% of patients in this study have a clear history of PA exposure. The remaining 19% of patients with PA-HSOS were unable to collect a clear PA intake history due to the following reasons: First, the patient had an unknown history of Chinese herbal intake, which has not yet been reported to contain PA. Secondly, when the clinicians collected the medical history, they did not give the relevant herb tips to the patient, so the possible PA intake history may be ignored. For patients without a clear history of PA intake, we classify them as BCS group if hepatic vein occlusion is detected by imaging methods. On the other hand, it is recognized that BCS-HV is often complicated by thrombophilia, the main causes of which are underlying hereditary and acquired hematological disorders [26,27]. The younger age of onset in patients with hematologic diseases may partly explain why BCS-HV patients are generally younger on average. Patients in the PA-HSOS group had a more rapid onset with a median time from symptom onset to the first presentation of one month; this was significantly shorter than in the BCS-HV group. This time frame parallels that which is seen in drug-induced liver disease [28,29]. In contrast, most splanchnic vein thromboses progress relatively slowly, leading to insidious symptoms and delayed diagnosis. A small number of patients in both groups developed abdominal wall varices and lower extremity edema, which were mainly associated with compression of the inferior vena cava by an enlarged liver.

Although BCS-HV and PA-HSOS patients present with abdominal pain, abdominal distention, ascites, jaundice, and lower limb edema, as reported in previous studies [16], we found that these symptoms were usually more severe or pronounced in patients with PA-HSOS. Similarly, in patients with PA-HSOS, laboratory tests such as liver, renal, and coagulation function were also worse than in BCS-HV patients. BCS patients may have a lower white blood cell count, serum hemoglobin concentration, and platelet count due to compensation that occurs with chronic disease [30]. We were unable to explain why platelet counts were lower in PA-HSOS than in BCS-HV patients. The increased hemoglobin in PA-HSOS patients could stem from hemoconcentration due to frequent large-volume paracentesis or the administration of diuretic drugs [13,31].

Both BCS-HV and PA-HSOS can manifest ascites, hepatomegaly, heterogeneous density, and plaque-like or map-like enhancements on CT or MRI imaging. Although there was a significant difference between the two groups in these characteristics, we cannot recommend them as specific distinguishing features without extensive validation work. We also found a significant difference between the two groups in the prevalence of venous thrombosis. Patients with BCS-HV showed a high incidence of mainly hepatic venous thrombosis, which is consistent with the BCS risk factors identified in our study. Conversely, the percentage of portal vein thrombosis in PA-HSOS patients was higher than in BCS-HV. This may be related to the significantly slower portal blood flow velocity in patients with PA-HSOS compared to BCS-HV; some studies have found that slower portal vein blood flow is more conducive to the formation of portal vein thrombosis [32,33].

Notably, patients with BCS-HV had a significantly higher incidence of localized stenosis or occlusion of the hepatic veins (100.00% vs. 1.20%), collateral circulation of hepatic veins (73.9% vs. 0), enlargement of the caudate lobe of the liver (47.7% vs. 0), and early enhancing nodules (8.46% vs. 0) than in PA-HSOS where these features were almost absent. Therefore, we believe that these four indicators may play an important role in differentiating these two diseases, although the lower incidence of caudate lobe hypertrophy and early enhancing nodules may hamper their sensitivity. Further, we focused on the differences between ultrasound and enhanced CT or MRI in distinguishing the above four indicators. Finally, we found that ultrasound can better display the local stenosis and occlusion of hepatic veins and the communicating branches between hepatic veins than enhanced CT or MRI. On the other hand, enhanced CT or MRI has advantages in finding enlargement of the caudate lobe of the liver and early liver nodule enhancement.

It is worth noting that, based on the huge difference in detection rate of patchy liver enhancement and heterogeneous hypoattenuation on enhanced CT (92.96% vs. 28.07%, 100% vs. 19.30%, respectively), Y Song et al. concluded that these two image features were important in distinguishing PA-HSOS from BCS. While in our study, although the detection rates of the two image features are different, the difference in patchy liver enhancement is much lower than Y Song’s study (92.66% vs. 78.46%). We speculate that the differences are likely due to the different definitions of patients with BCS included in the two studies, and the inclusion of BCS-IVC and BCS-C patients may reduce the detection rate of patchy liver enhancement in BCS patients, leading to a big difference from PA-HSOS patients. Therefore, using this image feature to distinguish BCS-HV and PA-HSOS will increase the false positive rate of PA-HSOS.

The importance of the significant differences between ultrasound and enhanced CT or MRI in identifying these important image features of BCS-HV and PA-HSOS may be as follows. First of all, the stricture or obstruction of the hepatic vein outflow tract and the communicating branch between hepatic veins are the most important features of BCS-HV, but the low detection rate of these features by enhanced CT or MRI may be an important reason for the misdiagnosis of BCS-HV as PA-HSOS, if ultrasound findings are ignored or unavailable. Second, although ultrasound can better identify the two features of BCS-HV established in our study, in clinical practice, ultrasound physicians can demonstrate very different abilities in diagnosing BCS-HV and PA-HSOS because these two diseases are rare. Without special training or particular instructions by the primary treating physician before the examination, these important features may be missed, leading to an improper diagnosis. Third, although enhanced CT or MRI is superior to ultrasound in finding caudate lobe enlargement and intrahepatic enhanced nodules, their clinical value is hampered by the relatively low incidence of these two features.

The main advantage of our work is that it involves a multicenter, large sample size study in which we conducted a comprehensive and systematic comparison of two rare hepatic vascular diseases using real-world clinical data. This study is the first to focus on comparing the hepatic venous occlusion type of Budd–Chiari syndrome with PA-HSOS, providing critical clinical indicators for clinicians to assist the identification of the two diseases. This study shows for the first time that there are significant differences between ultrasound and enhanced CT or MRI in the recognition of important image features in the two diseases, with a clear impact on the accuracy of the final diagnosis. Specifically, we emphasize a novel finding of the critical role of Doppler ultrasound in distinguishing BCS-HV from PA-HSOS, over enhanced CT or MRI.

Limitations of the study include its retrospective nature, which can increase the risk of selection bias, and that some patients in the cohort had incomplete data. Although we have some pathological specimens from PA-HSOS patients obtained by transjugular liver biopsy, this study did not compare the pathological features of the two diseases due to the lack of pathological data on BCS-HV, because liver biopsy is not routinely recommended for the diagnosis of BCS. Another disadvantage is that only a small percentage of BCS-HV patients had etiological data, so we ultimately did not analyze and present it in our study.

## 5. Conclusions

History of PA exposure, local stenosis, or occlusion of the hepatic veins, hepatic venous collateral circulation, hepatic caudate lobe enlargement, and early nodule enhancement in the liver, show important differential diagnostic value. Critically, we found that DUS is superior to enhanced CT or MRI in visualizing localized occlusions of the hepatic veins and collateral circulation of hepatic veins.

## Figures and Tables

**Figure 1 jpm-13-00603-f001:**
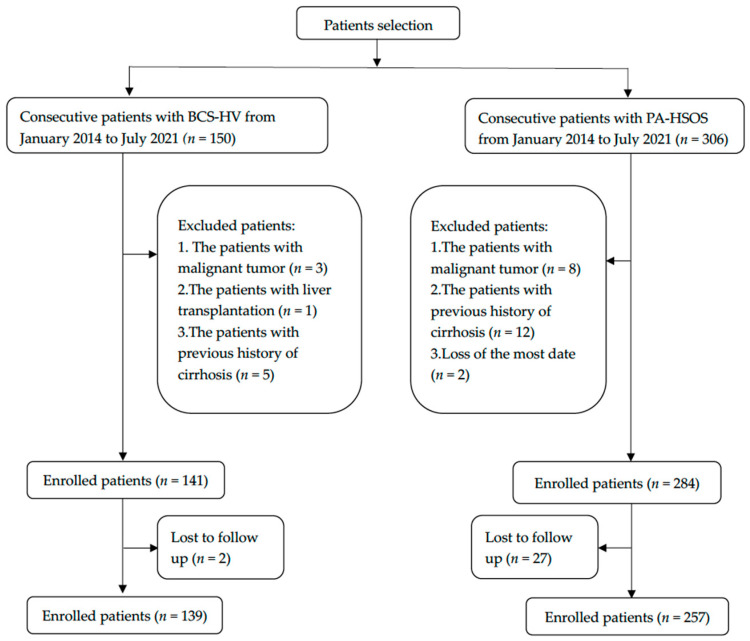
Flowchart of the enrollment of patients with BCS-HV or PA-HSOS.

**Figure 2 jpm-13-00603-f002:**
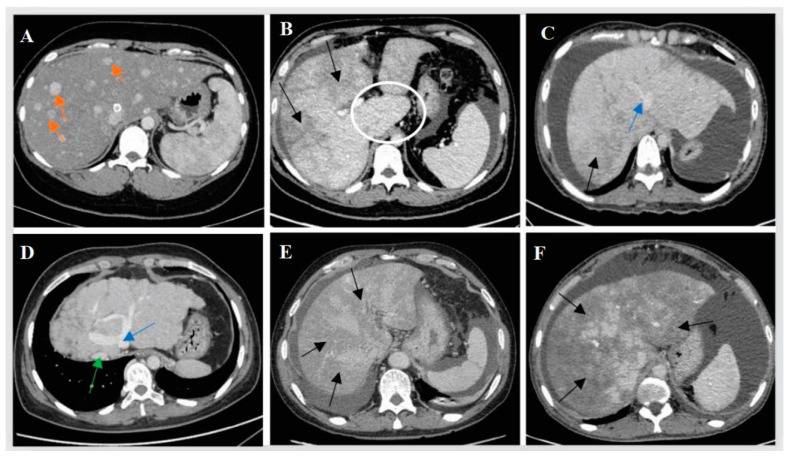
Patients diagnosed with BCS-HV and PA-HSOS underwent contrast-enhanced CT scans. (**A**–**D**): images of BCS-HV patients; (**E**,**F**): images of PA-HSOS patients; liver patchy enhancement and inhomogeneous hypodensity (black arrows); caudate lobe enlargement (white circles); early enhancing nodules (orange arrows); Enlarged hepatic vein (blue arrow); inferior vena cava (green arrow). (**A**): Multiple enhancing nodules in the parenchymal phase of the liver; (**B**): Heterogeneous patchy enhancement and enlargement of the caudate lobe; (**C**): Heterogeneous enhancement of the liver and dilated middle hepatic vein (blue arrows); (**D**): Collateral circulation among hepatic veins, narrowed inferior vena, and disconnected hepatic vein and inferior vena cava; (**E**,**F**): No liver nodules with early enhancement, no enlarged spleen and unevenly enhanced liver with a clover shape.

**Figure 3 jpm-13-00603-f003:**
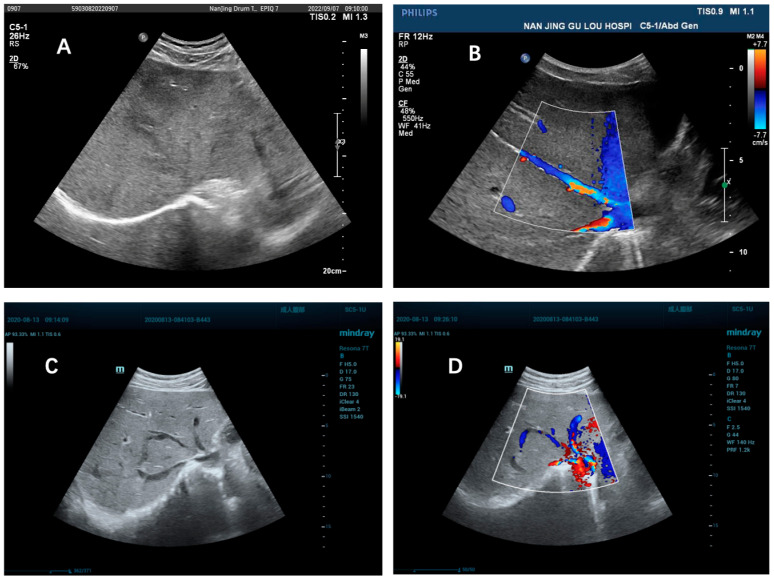
Ultrasound images of BCS-HV and PA-HSOS patients. (**A**) Uneven attenuation of the liver in a patient with PA-HSOS; (**B**) Unobstructed hepatic veins of the PA-HSOS patient; (**C**) Middle hepatic vein occlusion in BCS-HV patients; (**D**) Ultrasound images showed that the middle hepatic vein was occluded and collateral circulation of hepatic veins.

**Table 1 jpm-13-00603-t001:** The epidemiology and clinical manifestations of the patients.

Variables	BCS-HV (*n* = 139)	PA-HSOS (*n* = 257)	*p* Value
Age (year, mean ± SD)	43.00 ± 13.08	62.75 ± 9.68	<0.001
Male, *n* (%)	67 (48.20)	132 (51.40)	0.549
Course of disease (months, m (Q1~Q3))	3 (1.00–24.00)	1.00 (0.67–2.00)	<0.001
≤6 months, *n* (%)	85 (61.20)	249 (96.90)	<0.001
Abdominal distension, *n* (%)	89 (64.00)	257 (100)	<0.001
Abdominal pain, *n* (%)	24 (17.30)	68 (26.10)	0.047
Ascites	97 (69.78)	256 (99.60)	<0.001
None, *n* (%)	43 (31.20)	1 (0.40)	<0.001
Mild, *n* (%)	37 (26.80)	11 (4.30)	<0.001
Moderate to severe, *n* (%)	58 (42.00)	245 (95.30)	<0.001
Varices of abdominal wall, *n* (%)	13 (9.40)	11 (4.30)	0.044
Jaundice, *n* (%)	37 (26.60)	115 (44.70)	<0.001
Lower extremity edema, *n* (%)	28 (20.10)	66 (25.70)	0.217
History of PA intake, *n* (%)	0	210(81.71)	<0.001

Notes: Disease course: Time from symptom onset to first presentation.

**Table 2 jpm-13-00603-t002:** Baseline characteristics of the laboratory tests.

Variables	BCS-HV (*n* = 139)	PA-HSOS (*n* = 257)	*p* Value
WBC count, (×10^9^/L, m (Q1~Q3))	4.70 (3.20–6.32)	6.00 (4.80–7.90)	<0.001
HB (g/L, mean ± SD)	120.14 ± 29.92	143.78 ± 20.13	<0.001
PLT count, (×10^12^/L, m (Q1~Q3))	117.00 (78.00–185.00)	101.00 (72.00–134.50)	0.001
ALT (U/L, m (Q1~Q3))	23.00 (16.00–32.00)	48.80 (25.15–101.85)	<0.001
AST (U/L, m (Q1~Q3))	28.00 (21.00–40.00)	64.10 (40.00–107.45)	<0.001
ALP (U/L, m (Q1~Q3))	111.00 (78.00–149.00)	128.80 (97.40–174.05)	0.001
Albumin (g/L, mean ± SD)	36.45 ± 5.64	32.70 ± 3.52	<0.001
TBil (μmoI/L, m (Q1~Q3))	26.20 (16.06–45.33)	37.60 (25.40–55.65)	<0.001
SCr (μmoI/L, m (Q1~Q3))	57.00 (50.50–68.60)	72.00 (59.00–88.50)	<0.001
PT (s, m (Q1~Q3))	13.90 (12.50–15.90)	15.20 (13.70–17.55)	<0.001
APTT (s, m (Q1~Q3))	38.70 (31.50–62.00)	35.00 (30.15–43.95)	0.001
INR (m (Q1~Q3))	1.21 (1.09–1.37)	1.31 (1.19–1.52)	<0.001
Fib (g/L, m (Q1~Q3))	2.42 (2.00–2.86)	2.10 (1.70–2.70)	<0.001
Child–Pugh scores (mean ± SD)	7.43 ± 1.98	9.15 ± 1.48	<0.001
Child–Pugh class			
A, *n* (%)	51 (36.70%)	2 (0.80%)	
B, *n* (%)	65 (46.80%)	152 (59.10%)	
C, *n* (%)	23 (16.50%)	103 (40.10%)	

Notes: WBC, white blood cell; HB, hemoglobin; PLT, platelet count; ALT, alanine aminotransferase; AST, aspartate aminotransferase; ALP, alkaline phosphatase; TBil, total bilirubin; Scr, serum creatinine; PT, prothrombin time; APTT, activated partial thromboplastin time; Fib, fibrinogen; INR, international normalized ratio.

**Table 3 jpm-13-00603-t003:** Imaging features of patients with BCS-HV and PA-HSOS.

Variables	BCS-HV	PA-HSOS	*p* Value
Portal vein thrombosis, % (*n*/N)	4.4% (6/136)	11.70%(30/257)	0.018
Localized stenosis or occlusion of the hepatic veins, % (*n*/N)	100.00% (130/130)	1.20% (3/256)	<0.001
Imaging manifestations of cirrhosis, % (*n*/N)	47.10% (65/138)	2.00% (5/256)	<0.001
Hepatomegaly, % (*n*/N)	56.20% (73/130)	67.10% (163/243)	0.037
Splenomegaly, % (*n*/N)	63.80% (78/130)	16.10% (39/242)	<0.001
Collateral circulation of hepatic veins, % (*n*/N)	73.90% (99/134)	0	<0.001
Diameter of left hepatic vein, (cm, mean ± SD, *n*/N)	0.70 (0.50~1.09) (78/134)	0.39 (0.30~0.45) (156/160)	<0.001
Diameter of middle hepatic vein, (cm, mean ± SD, *n*/N)	0.70 (0.50~1.10) (70/134)	0.42 (0.35~0.49) (160/160)	<0.001
Diameter of right hepatic vein, (cm, mean ± SD, *n*/N)	0.80 (0.50~1.10) (54/134)	0.42 (0.36~0.49) (159/160)	<0.001
Diameter of portal vein, (cm, mean ± SD, *n*/N)	1.19 ± 0.27 (112/136)	1.00 ± 0.16 (243/257)	<0.001
Portal vein blood flow velocity, (cm/s, mean ± SD, *n*/N)	21.84 ± 9.22 (54/136)	15.74 ± 7.99 (238/238)	<0.001
Diameter of splenic vein, (cm, mean ± SD, *n*/N)	0.89 ± 0.29 (72/136)	0.60 ± 0.12 (230/230)	0.002
Splenic vein blood flow velocity, (cm/s, mean ± SD, *n*/N)	18.78 ± 9.42 (30/136)	13.47 ± 6.26 (230/230)	0.001
Patchy liver enhancement, % (*n*/N)	78.46% (102/130)	92.66% (164/177)	<0.001
Enlarged caudate lobe of the liver, *n*/N (%)	47.70% (62/130)	0	<0.001
Early strengthening nodules in the liver, *n*/N (%)	8.50% (11/129)	0	<0.001
Narrowed inferior vena cava, *n*/N (%)	55.20% (69/125)	44.20% (80/177)	0.087
Esophageal varicose veins, *n*/N (%)	64.00% (80/125)	21.20% (38/179)	<0.001

Notes: N: the total number of patients screened by related indicators; *n*: the number of individuals with positive indicators or screening results; Percentage (%): *n*/N.

**Table 4 jpm-13-00603-t004:** Comparison of color Doppler ultrasound and enhanced CT in BCS-HV and PA-HSOS.

Variables	BCS-HV			PA-HSOS		
	DUS	CT/MRI	*p* Value	DUS	CT/MRI	*p* Value
Hepatic venous stenosis or occlusion, % (*n*/N)	86.29% (107/124)	4.55% (5/110)	<0.001	1.88% (3/160)	2.83% (5/177)	0.726
Unclear display of hepatic vein, % (*n*/N)	12.90% (16/124)	94.55% (104/110)	<0.001	0.62% (1/160)	97.18% (172/177)	<0.001
Collateral circulation of hepatic veins, % (*n*/N)	70.97% (88/124)	4.55% (5/110)	<0.001	0	0	/
Enlarged caudate lobe, % (*n*/N)	/	47.27% (52/110)	/	/	0	/
Early strengthening nodules, % (*n*/N)	/	8.18% (9/110)	/	/	0	/

Notes: Comparison of color Doppler ultrasound and enhanced CT in two groups. CT: computed tomography; DUS: Doppler ultrasound; N: the total number of patients screened by related indicators; *n*: the number of individuals with positive indicators or screening results; Percentage (%): *n*/N.

## Data Availability

Data sets used and/or analyzed in the current study may be obtained from the corresponding author upon reasonable request.

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
