# Peer review of "Hepatic Venous Occlusion Type of Budd–Chiari Syndrome versus Pyrrolizidine Alkaloid-Induced Hepatic Sinusoidal Obstructive Syndrome: A Multi-Center Retrospective Study"

_jpm, 2023, doi:10.3390/jpm13040603_

Round 1

Reviewer 1 Report

The authors attempt to address a challenging clinical entity, distinguishing intra-hepatic venous obstruction from outflow insufficiency.

Some major shortcomings include overlap in the definitions of the two groups (missing criteria of PA ingestion). Also, the study design is not well represented, such that it is difficult to discern how the authors addressed some of these difficulties.

Title: recommend against using abbreviations "BC", "PA", "HSOS"

Abstract: there are too many abbreviations to understand the message. 

Methods: The study design is not well-described. The patients are noted to be enrolled retrospectively, but it is not clear if the CT and MRI were repeated as part of the study. Also, was the ultrasound conducted as part of the study or the data were collected retrospectively.

It is not clear what is meant by "traffic branch". I cannot find this term in the literature. This term is used throughout the paper.

Results: p value cannot be zero.  this is present throughout the paper.

If a history of taking PA intake was necessary for diagnosis of PA-HSOS, why do only 81% of PA-HSOS have a history of PA intake? How was BC-HV ruled out in these cases given the noted similarity on presentation and cross sectional imaging?

Although the difference in platelets is reported as not significant, the p value is <0.01 in the table.

Figure 1: Please clarify what is meant by "3. loss of the most date"

Table 1: please clarify if these are means or medians. the numbers in parentheses seem to be a range, is it the total range or the IQR?

imaging findings: this is the first time DSA is mentioned. please include in methods when this was employed and by whom: the study investigators or the treating physicians.

Cirrhosis is not an imaging finding. Unless biopsy information is provided, it is not appropriate to label a subject with the diagnosis of cirrhosis. if nodular contour is the finding of interest, then report as such.

Given that stenosis is a major finding, a definition of how it was defined would be beneficial. I assume there are different criteria on CT/MRI and US.

Tables 3, 4. It is not clear which imaging modalities are included in table 3. For instance, Hepatic stenosis/occlusion was found in 100% of BCS-HV, but table 4 reports 86% by ultrasound and 4.6% by CT/MRI. These findings seem to be in direct conflict of each other.

Author Response

Reviewer(s)' Comments to Author:

The authors attempt to address a challenging clinical entity, distinguishing intra-hepatic venous obstruction from outflow insufficiency.

Some major shortcomings include overlap in the definitions of the two groups (missing criteria of PA ingestion). Also, the study design is not well represented, such that it is difficult to discern how the authors addressed some of these difficulties.

Response: Thanks for your great suggestion. We have added a description of PA intake in the revised article. The patient intake some plants containing PA reported in previous researches, such as panax Gynura japonica,which is considered to have a clear history of PA administration.

Title: recommend against using abbreviations "BC", "PA", "HSOS"

Response: Thanks again for your kind information and great suggestion. We have modified the title of the article and deleted the abbreviations.

Abstract: there are too many abbreviations to understand the message.

Response: Thanks for your kind suggestion. We have revised the article summary and deleted the abbreviations for better understanding and reading.

Methods: The study design is not well-described. The patients are noted to be enrolled retrospectively, but it is not clear if the CT and MRI were repeated as part of the study. Also, was the ultrasound conducted as part of the study or the data were collected retrospectively.

Response: Thanks for your suggestion. We made a supplementary explanation in the method section. All imaging data were collected retrospectively.

It is not clear what is meant by "traffic branch". I cannot find this term in the literature. This term is used throughout the paper.

Response: Thanks you for your valuable suggestion. All "traffic branch " words in the article have been modified to "collateral circulation of the hepatic veins".

Results: p value cannot be zero.  this is present throughout the paper.

Response: Thanks, great suggestion! All parts with p value of 0 in the article have been modified to "<0.001".

If a history of taking PA intake was necessary for diagnosis of PA-HSOS, why do only 81% of PA-HSOS have a history of PA intake? How was BC-HV ruled out in these cases given the noted similarity on presentation and cross sectional imaging?

Response: Thanks you for your valuable suggestion. The diagnosis of these patients depends on the clinical manifestations and imaging features, especially the acute course of disease, abdominal distension, decreased hepatic venous flow rate, decreased liver uneven density, no previous history of special liver disease, and a history of unknown Chinese herbal medicine. The remaining 19% of patients with PA-HSOS were unable to collect a clear PA intake history due to the following reasons: First, the patient had an unknown history of Chinese herbal intake, which has not yet been reported to contain PA. Secondly, when the clinicians collected the medical history, they did not give the relevant herb tips to the patient, so the possible PA intake history may be ignored.

For patients without a clear history of PA intake,we classify them as BCS group if hepatic vein occlusion is detected by imaging methods.

Although the difference in platelets is reported as not significant, the p value is <0.01 in the table.

Response: Thanks, the comparison of platelet levels has been modified in the article.

Figure 1: Please clarify what is meant by "3. loss of the most date"

Response: Thanks, "3. Loss of the most date" means the loss of most laboratory examination and imaging data, which has been added in the article.

Table 1: please clarify if these are means or medians. the numbers in parentheses seem to be a range, is it the total range or the IQR?

Response: Thanks for your advice. Whether the data is median, mean or quartile has been further indicated in the tables.

imaging findings: this is the first time DSA is mentioned. please include in methods when this was employed and by whom: the study investigators or the treating physicians.

Response: Thanks for your suggestion. The specific operation of DSA has been supplemented by the operator in the method section.

Cirrhosis is not an imaging finding. Unless biopsy information is provided, it is not appropriate to label a subject with the diagnosis of cirrhosis. if nodular contour is the finding of interest, then report as such.

Response: Thanks for your suggestion. We added the definition of cirrhosis in the method section of the article. The diagnostic criteria of liver cirrhosis in this study are as follows: â‘  the appearance of liver is not smooth; â‘¡ Liver fissure is widened; â‘¢ The patient who has received liver biopsy in the past was confirmed as cirrhosis.

Given that stenosis is a major finding, a definition of how it was defined would be beneficial. I assume there are different criteria on CT/MRI and US.

Response: Thanks for your suggestion. In the method part of the article, we further supplemented the definition of local hepatic vein stenosis or occlusion, including the difference between DUS and enhanced CT or MRI.

Tables 3, 4. It is not clear which imaging modalities are included in table 3. For instance, Hepatic stenosis/occlusion was found in 100% of BCS-HV, but table 4 reports 86% by ultrasound and 4.6% by CT/MRI. These findings seem to be in direct conflict of each other.

Response: Thanks for your suggestion. Thanks for your suggestion. The collection methods of patient imaging data recorded in Table 3 include all methods which include ultrasound, enhanced CT, MRI, and DSA. If any one of these methods identified hepatic venous stenosis or occlusion,BCH-HV can be diagnosed. In addition, some patients lack ultrasound, enhanced CT, or MRI data. Therefore, the results in Table 3 and Table 4 are not consistent.

Reviewer 2 Report

The authors are presenting their research results regarding potential differential characteristics of PA-HSOS and HV-BCS, based on a multicenter patient cohort. They conclude that clinical features, including abdominal distension, abdominal pain, ascites, abdominal varicose veins, and jaundice, overlapped between the two groups, with significantly different incidence, also laboratory parameters were more deranged in the PA-HSOS group. The study focuses also on imagistic diagnostic modalities, comparing the sensitivity of the CT and MRI scan with the Doppler ultrasound investigation.

As far as I can appreciate it, the results are statistically relevant.  

To my knowledge, there are only few publications covering the subject and, in regions where the PA-HSOS is frequent, an exact differentiation between the two liver pathologies can be of great importance for conducting the patient’s therapy.

The fact that Doppler ultrasound is described to be more sensitive than CT or MRI scans is of great importance, when considering the cost burden and potential side effects of imaging. 

The review is in my opinion adequately covering the subject, the pillars of diagnosis being discussed and graded.

The manuscript is clear and well structured, the language adequate and easy to read, the derived conclusions are in logical line with the argumentation. The cited references are relevant to the topic. The tables are clear and easy to understand. 

The conclusions are supported by the listed citations.

The work is no novelty on the field but is adding valuable information regarding the differentiation of two rare entities which could be missed or confused.  

I suggest revising the third literature note and deleting the number 3 which is most probably a typing error.  I may add additional value to the paper if the discussion would contain a reference to the therapeutic relevance of the accurate diagnosis of these two conditions and the general management of this different groups of patients.  

Author Response

Response: Thank you very much for your evaluation of this study. We are committed to improving the diagnosis of BCS-HV and HSOS caused by pyrrole alkaloids and have drawn valuable conclusions. The third literature note has been modified.

Round 2

Reviewer 1 Report

Comments have been adequately addressed.